# Expanding the *Drosophila* toolkit for dual control of gene expression

Jonathan Zirin[1], Barbara Jusiak[2†], Raphael Lopes[1†], Benjamin Ewen-Campen[1], Justin A Bosch[1], Alexandria Risbeck[1], Corey Forman[1], Christians Villalta[1], Yanhui Hu[1], Norbert Perrimon[1,3*]

[1]Department of Genetics, Harvard Medical School, Boston, United States; [2]Department of Physiology and Biophysics, University of California, Irvine, Irvine, United States; [3]Howard Hughes Medical Institute, Boston, United States

*For correspondence:
perrimon@genetics.med.
harvard.edu

†These authors contributed
equally to this work

Reviewing Editor: Claude
Desplan, New York University,
United States

**Abstract** The ability to independently control gene expression in two different tissues in the same animal is emerging as a major need, especially in the context of inter-organ communication studies. This type of study is made possible by technologies combining the GAL4/UAS and a second binary expression system such as the LexA system or QF system. Here, we describe a resource of reagents that facilitate combined use of the GAL4/UAS and a second binary system in various *Drosophila* tissues. Focusing on genes with well-characterized GAL4 expression patterns, we generated a set of more than 40 LexA-GAD and QF2 insertions by CRISPR knock-in and verified their tissue specificity in larvae. We also built constructs that encode QF2 and LexA-GAD transcription factors in a single vector. Following successful integration of this construct into the fly genome, FLP/FRT recombination is used to isolate fly lines that express only QF2 or LexA-GAD. Finally, using new compatible shRNA vectors, we evaluated both LexA and QF systems for in vivo gene knockdown and are generating a library of such RNAi fly lines as a community resource. Together, these LexA and QF system vectors and fly lines will provide a new set of tools for researchers who need to activate or repress two different genes in an orthogonal manner in the same animal.

## eLife assessment

This **important** study reports the generation of genetic tools for manipulating several tissues at the same time in *Drosophila*. The authors provide **convincing** evidence that this allows for the generation of LexA and QF2 driver lines, which will be of great utility for understanding inter-organ communication. Making the tools available through the *Drosophila* stock center and plasmid depository will ensure that they are easily accessed by many researchers.

## Introduction

### Combinatorial binary systems

Most reagents available for loss-of-function (LOF) and gain-of-function studies using RNAi or CRISPR rely on GAL4/UAS-mediated expression (*Brand and Perrimon, 1993*; *Dietzl et al., 2007*; *Perkins et al., 2015*; *Zirin et al., 2020*; *Port and Boutros, 2022*). However, some studies, such as the study of intercellular or inter-organ communication, require the simultaneous use of two independent binary transcriptional systems. For example, dual expression systems have been used to study how a *Drosophila* insulin-like peptide released from the wing primordium communicates with the brain to control organ growth (*Colombani et al., 2015*), analyze signaling from olfactory neurons to blood cells (*Shim et al., 2013*), independently manipulate ligand-producing and ligand receiving cells (*Yagi et al., 2010*), and visualize interactions between clonal cell populations in tissues (*Bosch et al., 2015*).

**eLife digest** In order for researchers to understand how organisms develop and function, they often switch specific genes on or off in certain tissues or at selected times. This can be achieved using genetic tools called binary expression systems. In the fruit fly – a popular organism for studying biological processes – the most common is the GAL4/UAS system.

In this system, a protein called GAL4 is expressed in a specific organ or tissue where it activates a UAS element – a genetic sequence that is inserted in front of the gene that is to be switched on. This can also include genes inserted into the fruit fly encoding fluorescent proteins or stretches of DNA coding for factors that can silence specific genes. For example, fruit flies expressing GAL4 protein specifically in nerve cells and a UAS element in front of a gene for a fluorescent protein will display fluorescent nerve cells, which can then be examined using fluorescence microscopy.

Studying how organs communicate with one other can require controlled expression of multiple genes at the same time. In fruit flies, other binary expression systems that are analogous to the GAL4/UAS system (known as LexA/LexAop and QF/QUAS) can be used in tandem. For example, to study gut-brain communication, the GAL4/UAS system might be used to switch on the gene for an insulin-like protein in the gut, with one of the other systems controlling the expression of its corresponding receptor in the brain. However, these experiments are currently difficult because, while there are thousands of GAL4/UAS genetic lines, there are only a few LexA/LexAop and QF/QUAS genetic lines.

To address this lack of resources, Zirin et al. produced a range of genetically engineered fruit flies containing the LexA/LexAop and QF/QUAS binary expression systems. The flies expressed LexA or QF in each of the major fly organs, including the brain, heart, muscles, and gut. A fluorescent reporter gene linked to the LexAop or QUAS elements, respectively, was then used to test the specificity to single organs and compare the different systems. In some organs the LexA/LexAop system was more reliable than the QF/QUAS system. However, both systems could be successfully combined with genetic elements to switch on a fluorescent reporter gene or switch off a gene of interest in the intended organ.

The resources developed by Zirin et al. expand the toolkit for studying fruit fly biology. In future, it will be important to understand the differences between GAL4, LexA and QF systems, and to increase the number of fruit fly lines containing the newer binary expression systems.

---

Based on the need to simultaneously manipulate different sets of cells in a given tissue, the LexA/LexAop system (*Lai and Lee, 2006*) and the QF/QUAS system (*Potter et al., 2010*; *Potter and Luo, 2011*) have been developed. There have been no systematic studies comparing the two systems, with only anecdotal evidence to support one system over the other.

The numbers of available LexA and QF fly lines with tissue-specific expression domains are far lower than that represented by the thousands of GAL4 enhancer lines, which have been developed using various approaches over the past 25 years. The largest existing set of LexA system driver lines were produced by the Janelia FlyLight Project and Vienna Tiles Project (*Jenett et al., 2012*; *Tirian and Dickson, 2017*). However, these lines were developed primarily for nervous system expression. Although they are often expressed in other tissues, they are not well suited for experiments targeting non-neuronal cell types. Furthermore, the FlyLight lines use a p65 transcriptional activation domain and therefore are not compatible with the GAL80 temperature-sensitive GAL4 repression system. A second large collection of ~180 LexA-based enhancer trap fly stocks has been generated (the StanEx collection) (*Kockel et al., 2016*; *Kockel et al., 2019*; *Kim et al., 2023*). On average, each StanEx line expresses LexA activity in five distinct cell types, with only one line showing expression in just one tissue, unfortunately limiting usefulness of these reagents (*Kockel et al., 2016*). These findings are consistent with prior studies indicating that enhancers very rarely produce expression patterns that are limited to a single cell type in a complex organism (*Jenett et al., 2012*). Regarding the Q system, there are 101 total QF lines available from the Bloomington *Drosophila* Stock Center (BDSC). As the original QF can be toxic when expressed at high levels (*Potter et al., 2010*), second-generation QF2 and QF2w, which are much less toxic and can be expressed broadly in vivo, were generated (*Riabinina et al., 2015*). Among the 51 QF2 and QF2w lines available at BDSC, most are expressed in the brain, with relatively few of these drivers expressed specifically in other tissues. Thus, there remains an

unmet need for more LexA and QF drivers with tissue-specific expression patterns. Furthermore, there are only ~260 LexA driver-compatible LexAop and ~130 QF driver-compatible QUAS stocks available at BDSC. The vast majority of these are to induce expression of fluorescent reporter genes, rather than molecular genetic reagents such as shRNAs for RNAi. This lack of fly stock reagents dramatically slows down studies that require two independent binary transcriptional systems, as custom fly stock reagents must be made by individual groups.

## Rapid and efficient generation of driver lines by CRISPR-Cas9

Several methods have been developed to generate new drivers with well-established tissue-specific patterns. However, these methods require either de novo generation of new driver lines, as for the integrase swappable in vivo targeting element (InSITE) system (*Gohl et al., 2011*), or PhiC31-induced insertion of a transcription factor cassette into an existing minos-mediated integration cassette genomic insertion (*Venken et al., 2011*; *Diao et al., 2015*; *Gnerer et al., 2015*). CRISPR/Cas9 technology now makes it possible to knock-in a transcriptional activator into any locus. Some groups have also recently described tools to swap QF or LexA into the GAL4 coding region of existing GAL4 enhancer trap lines (*Lin and Potter, 2016*; *Chang et al., 2022*; *Karuparti et al., 2023*). The conversion can be performed through genetic crosses; however, the frequency of conversion can vary greatly among different GAL4 lines, and many such swaps do not fully reproduce the original GAL4 expression patterns (*Chen et al., 2019*).

Despite these techniques, and the growing collection of LexA-GAD and QF2 lines, progress has been slow, and the choice of tissues depends on the specific interests of individual labs. Here, we describe the efforts of the Transgenic RNAi Project (TRiP) to facilitate combinatorial studies by building reagents that can be used alongside GAL4/UAS in a variety of tissues. Focusing on genes with well-characterized GAL4 expression patterns, we targeted 23 genes to produce 44 new highly tissue-specific LexA-GAD and QF2 driver lines by CRISPR-mediated homology-directed repair. We chose to use LexA with the GAL4 activation domain, rather than the p65 or VP16 activation domains to allow for temporal control by the temperature-sensitive GAL4 repressor, GAL80 (*Lai and Lee, 2006*; *Pfeiffer et al., 2010*). We chose to use QF2 variant over the original QF, to avoid the toxicity reported for the latter (*Riabinina et al., 2015*). Like GAL80-based modulation of LexA-GAD, QF2 activity can also be regulated temporally by expressing QS, a QF repressor. QS repression of QF can be released by feeding flies quinic acid (*Riabinina and Potter, 2016*). Each knock-in was rigorously genotyped, and the expression pattern verified by imaging. Thus, we can systematically compare LexA-GAD vs QF2 activators inserted at precisely the same position in the genome. In addition to the new fly stocks, we provide a new set of vectors and protocols to efficiently generate LexA-GAD and/or QF2 drivers. This includes constructs that encode QF2 and LexA-GAD transcription factors in a single vector, with each coding sequence flanked by FRT sites. Following successful integration into the fly genome, the vector generates a driver in which a target gene expresses both QF2 and LexA-GAD. If desired, one of the two coding regions can then be excised with Flp, resulting in flies that express only QF2 or LexA-GAD. Furthermore, we evaluated both QF2 and LexA-GAD systems for in vivo gene knockdown and are generating a compatible library of transgenic shRNA lines in our custom QUAS and LexAop vectors as a community resource. Together, these QF2/LexA-GAD and QUAS/LexAop vectors and fly lines will provide a new set of tools for researchers who need to activate or repress two different genes in an orthogonal manner in the same animal.

## Results

### Strategy for CRISPR knock-in of T2A-LexA-GAD and T2A-QF2

We made each genetic driver line by inserting T2A-LexA-GAD or T2A-QF2 in the coding sequence of a target gene. We have previously reported the CRISPaint method to insert T2A-GAL4 into any gene using homology-independent repair (*Bosch et al., 2020*). Using this method, we showed robust gene-specific integration of donor plasmids in the fly germ line and successfully generated new driver lines. However, because of the high probability of indels at the insertion site, we opted to use traditional CRISPR homology-directed repair to insert the T2A-LexA-GAD and T2A-QF2 into the genome. We first modified the CRISPaint donor vectors to produce pHDR-T2A-LexA-GAD-Hsp70-3xP3-RFP and pHDR-T2A-QF2-Hsp70-3xP3-RFP, which contain the transcriptional activators followed by Hsp70

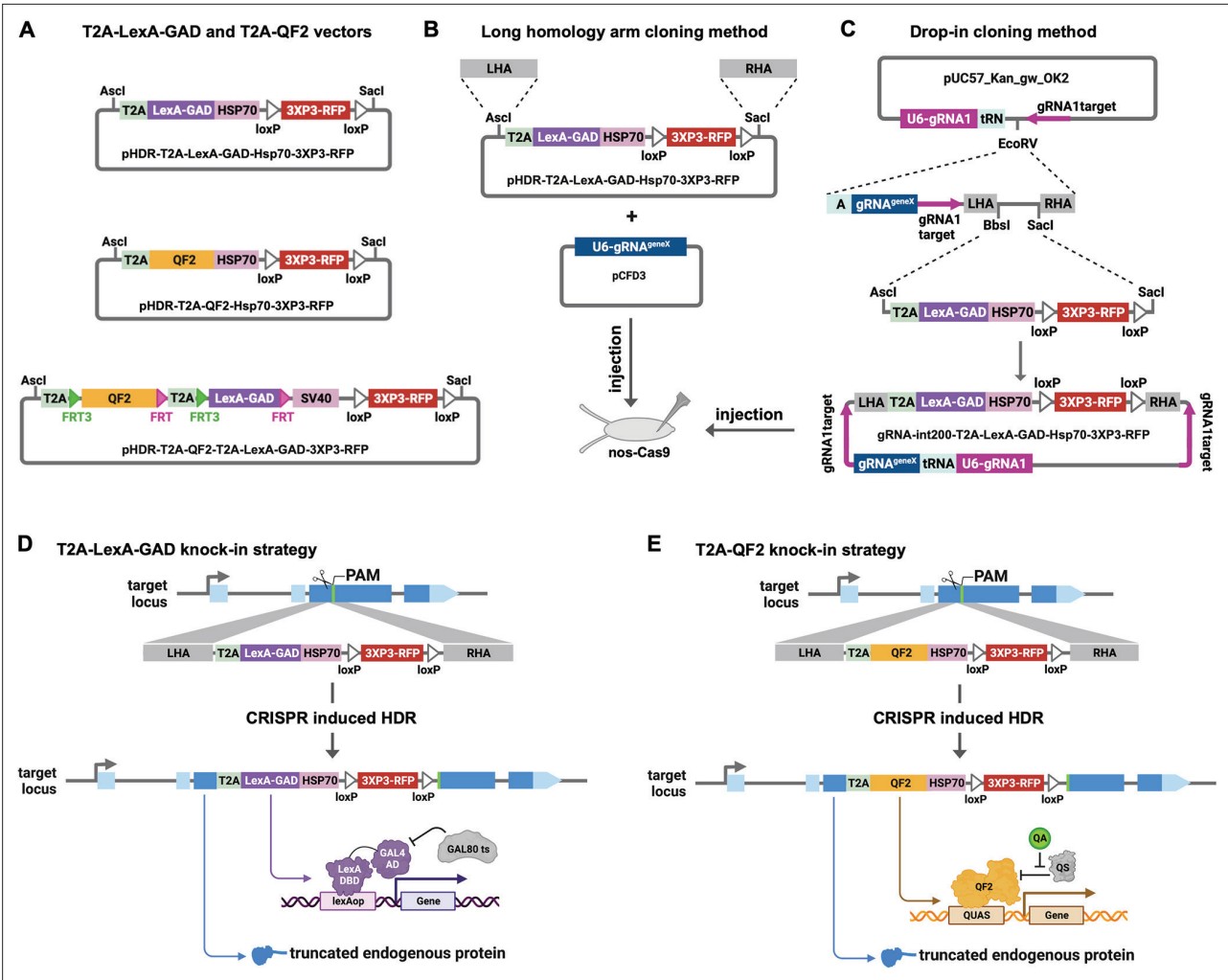

**Figure 1.** Strategy for CRISPR knock-in of T2A-LexA-GAD and T2A-QF2. (**A**) Donor vectors for knock-in. pHDR-T2A-LexA-GAD-Hsp70-3xP3-RFP and pHDR-T2A-QF2-Hsp70-3xP3-RFP contain T2A-LexA-GAD or QF2 transcriptional activators followed by HSP70 terminators. pHDR-T2A-QF2-T2A-LexA-GAD-3xP3-RFP contains both activators flanked by FRT sites followed by SV40 terminator. All the vectors contain a 3XP3-RFP transformation marker flanked by loxP sites. (**B**) Long homology arm cloning method. ~1000 bp homology arms are amplified from genomic DNA and inserted into the AscI and SacI sites by Gibson assembly. A separate target-gene-specific guide RNA is cloned into U6 promoter expression vector such as pCFD3. (**C**) Drop-in cloning method. Based on the gRNA-int200 method previously described (*Kanca et al., 2022*). A company synthesizes and clones a DNA fragment into the pUC57_Kan_gw_OK2 vector. The resulting plasmid contains the following elements: (1) two guide RNAs under the control of a U6 promoter, one (gRNA1) targeting the vector (pink arrows) to linearize the homology donor in vivo, and another (gRNA^geneX) targeting the gene of interest; (2) a tRNA sequence to allow liberation of the individual guides by the endogenous tRNA processing machinery; (3) 200 bp short homology arms; and (4) BbsI and SacI cloning sites. The AscI/SacI T2A-LexA-GAD, T2A-QF, or T2A-LexA-GAD-T2A-QF fragments can then be ligated in a single directional cloning step into the BbsI/SacI sites to produce the donor plasmid. (**D**) T2A-LexA-GAD and (**E**) T2A-QF2 knock-in strategy. CRISPR-based HDR causes integration of the T2A-LexA-GAD or T2A-QF2 in the most 5′ coding exon common to all or most isoforms, resulting in expression of the activators under control of the endogenous gene regulatory region. The knock-in also produces a truncated endogenous protein and thus a strong loss-of-function allele.

terminators. We opted to use the Hsp70 3′UTR, as opposed to SV40 3′UTR, because it is comparatively weaker and may avoid toxicity due to overexpression of LexA-GAD or QF2. We also constructed pHDR-T2A-QF2-T2A-LexA-GAD-3xP3-RFP, which contains both activators flanked by FRT sites followed by SV40 terminator (*Figure 1A*). All the vectors contain a 3XP3-RFP transformation marker gene flanked by loxP sites. These vectors are compatible with two different cloning methods for making a CRISPR donor plasmid. In the long homology arm (HA) method (*Figure 1B*), ~1000 bp HAs are amplified from genomic DNA and inserted into the donor plasmid by Gibson assembly. A separate target-gene-specific guide RNA is cloned into a U6 promoter expression vector such as pCFD3 (*Port et al., 2014*). In the 'drop-in' cloning method (*Figure 1C*; *Kanca et al., 2022*), a company synthesizes

**Table 1.** T2A-LexA-GAD and T2A-QF2 knock-in lines.

| Target tissue | Target gene (name) | QF2 | This paper | 3XP3 removed | Others at BDSC | LexA-GAD | This paper | 3XP3 removed | Others at BDSC |
|---|---|---|---|---|---|---|---|---|---|
| Ubiquitous | da | Yes | Yes | Yes | No | Yes | Yes | Yes | No |
| Germline | vas* | Yes | Yes | No | No | Yes | Yes | No | No |
| Muscle | Mef2 | Yes | Yes | Yes | 66469 | Yes | No | Yes | 97530 |
| Muscle | bt* | Yes | Yes | No | No | Yes | Yes | No | No |
| Pan-neuronal | elav† | Yes | Yes | Yes | 66466 | Yes | Yes | Yes | No |
| Glia | repo | Yes | Yes | No | 66477 | Yes | Yes | Yes | 97535 |
| Insulin-producing cells | Ilp2 (regulatory region)† | Yes | Yes | N/A | No | Yes | Yes | N/A | No |
| Fat | apolpp | Yes | Yes | Yes | No | Yes | Yes | Yes | No |
| Trachea | btl | Yes | Yes | Yes | No | Yes | Yes | Yes | No |
| Heart | Hand* | Yes | Yes | Yes | No | Yes | Yes | Yes | No |
| Enocyte | CG9458* | Yes | Yes | No | No | Yes | Yes | Yes | No |
| Enocyte | CG17560* | Yes | Yes | No | No | Yes | Yes | No | No |
| Salivary gland | Sgs3* | Yes | Yes | No | No | Yes | Yes | Yes | No |
| Prothoracic gland | phtm | Yes | Yes | Yes | No | Yes | Yes | Yes | No |
| Midgut | mex1 | Yes | Yes | Yes | No | Yes | Yes | Yes | No |
| Adult midgut enterocyte | Myo31DF | Yes | Yes | Yes | No | Yes | Yes | Yes | No |
| Hemocyte | Hml | Yes | Yes | Yes | 66468 | Yes | Yes | Yes | No |
| Hemocyte | He | Yes | Yes | Yes | No | No | No | N/A | No |
| Crystal cell | PPO1 | Yes | Yes | Yes | No | Yes | Yes | Yes | No |
| Lamellocyte | PPO3 | Yes | Yes | Yes | No | Yes | Yes | Yes | No |
| Posterior segment | hh† | Yes | Yes | Yes | No | Yes | Yes | Yes | 97536 |
| Wing pouch/hinge | nub | Yes | Yes | Yes | No | Yes | Yes | Yes | No |
| Imaginal disc A/P boundary | dpp (regulatory region)† | Yes | Yes | N/A | No | Yes | Yes | N/A | No |

*Constructs were cloned using the drop-in method. All others were cloned by PCR of long homology arms.
†Constructs were cloned into double driver vectors.

and clones a plasmid that contains all of the necessary features for CRISPR HDR plus a cloning site to allow ligation of T2A-LexA-GAD, T2A-QF, or T2A-LexA-GAD-T2A-QF fragments in a single step to produce the donor plasmid.

For our targets, we selected a set of genes with tissue-specific expression patterns encompassing most of the major organs of the fly (*Table 1*). When possible, we selected genes which had a previously characterized GAL4 insertion, and strong evidence of tissue specificity (e.g. from publicly available scRNAseq, in situ, and immunohistochemistry). Once the donors/guides were cloned, they were injected into Cas9-expressing fly embryos to induce CRISPR-based HDR of the T2A-LexA-GAD or T2A-QF2 (*Figure 1D and E*). We selected integration sites in the most 5' coding exon common to all or most isoforms, resulting in expression of the activators under control of the endogenous gene regulatory regions. The knock-in also produces a truncated endogenous protein and thus a predicted strong LOF allele. To verify the knock-ins, we PCR-amplified the genomic regions flanking the insertion sites and confirmed that the insertions were seamless and in-frame. Most of our knock-in stocks were made with the long HA method, but we transitioned to the drop-in method as this technology became available. As *Table 1* shows, we were able to successfully generate knock-ins into nearly all the target genes using these methods.

## Specificity of T2A-LexA-GAD and T2A-QF2 knock-in lines

Next, we tested the specificity of the knock-in driver lines in the third instar larva (*Figure 2*). Each T2A-LexA-GAD and T2A-QF2 knock-in line was crossed to a LexAop-GFP and QUAS-GFP reporter, respectively. Note that most of the lines are highly tissue-specific and are comparable between the LexA-GAD and QF2 knock-ins. Insertions in the *daughterless* gene *(da)* are an exception, as the T2A-LexA-GAD (*Figure 2A*), but not the T2A-QF2 (*Figure 2B*), gives the expected ubiquitous expression pattern. Similarly, for insertions in the *nubbin* gene *(nub)* the T2A-LexA-GAD (*Figure 2LL*), but not the T2A-QF2 (*Figure 2MM*), gives the expected expression in the wing imaginal disc. In both cases, T2A-QF is expressed in the correct tissues, but incompletely. Even with these exceptions, the patterns are remarkably consistent between the T2A-LexA-GAD and T2A-QF2 knock-in lines overall.

Our donor plasmids contain the transgenesis marker 3XP3-RFP, which expresses red fluorescence in the larval gut and anal pad, and the adult eye (*Berghammer et al., 1999*). Like our previous knock-ins using the CRISPaint method (*Bosch et al., 2020*), we sometimes observed LexA-Gad or QF2 expression in the larval anal pad and gut, coincident with expression of RFP. *Figure 3* shows one such example in the *repo* gene, where T2A-QF2 is expressed in the expected glial cells, but also misexpressed in RFP positive cells. Interestingly, this influence of the 3XP3-RFP is not observed in T2A-LexA-GAD *repo* knock-in animals (*Figure 2I*). Conversely, in the *breathless (btl)* gene, we observe misexpression of LexA-GAD, but not QF2 in RFP positive cells (*Figure 2M and N*). Importantly, when we removed the 3XP3-RFP cassette in the repo knock-in by cre-lox recombination, misexpression of QF2 in the gut and anal pad was completely eliminated, while the glial expression remained (*Figure 3B*). We therefore have removed or are in the process of removing the 3XP3-RFP marker from all the knock-in stocks. Currently, we have successfully removed it from 32 of the 41 driver lines that used this marker (*Table 1*).

## T2A-QF2-T2A-LexA-GAD double driver lines

To make the T2A-LexA-GAD or T2A-QF2 knock-ins described above, each donor plasmid must be individually cloned and injected into embryos. We reasoned that by combining both drivers into a single construct, we could halve the number of injections, one of the most labor-intensive and expensive parts of the process. For this combined expression of LexA-GAD and QF2 transcription factors, we built two types of vectors: (1) a CRISPR donor version, pHDR-T2A-QF2-T2A-LexA-GAD-3xP3-RFP, which is used to insert the cassette into an endogenous locus of interest (*Figure 4A*), and (2) a phiC31-attB version, pMCS-T2A-QF2-T2A-LexA-GAD-WALIUM20, which is used to clone an enhancer fragment of interest and is then integrated into an attP site in the fly's genome (*Figure 4B*). Note that the phiC31-attB-compatible constructs use mini-white, not 3XP3-RFP, as a marker gene. Using these vectors, we generated CRISPR knock-ins into the *elav* and *hedgehog (hh)* genes and enhancer lines for the *decapentaplegic (dpp)* and *insulin-like peptide 2 (Ilp2)* (*Figure 4F*) genes. The *elav* knock-in (*Figure 4C*) expressed both T2A-QF2 and T2A-LexA-GAD in the expected pattern in the larval nervous system. However, we did see some weak non-specific expression of T2A-LexA-GAD in the somatic muscles. The *hh* knock-in (*Figure 4D*) expressed both T2A-QF2 and T2A-LexA-GAD in the expected pattern in the posterior of the imaginal discs. However, we observed that T2A-QF2, but not T2A-LexA-GAD, was restricted from the wing pouch, similar to the individual *nub* knock-ins (*Figure 4D* and *Figure 2LL–MM*). As expected, the *dpp* enhancer (*Figure 4E*) directed expression of both T2A-QF2 and T2A-LexA-GAD along the anterior/posterior margin of the wing imaginal disc. Again, however, T2A-QF was much reduced in the wing pouch. Finally, the *Ilp2* enhancer (*Figure 4F*) directed expression of both T2A-QF2 and T2A-LexA-GAD very specifically in the insulin-producing cells of the larval brain. Taken together, these results show that the double driver constructs can effectively drive expression of both activators simultaneously in the target tissue, with the caveat that T2A-QF2 does not express well in the wing pouch.

Next, we attempted to derive single T2A-QF2 and T2A-LexA-GAD lines from T2A-QF2-T2A-LexA-GAD double drivers. The strategy is outlined in *Figure 5A*. FRT3 and FRT are mutually incompatible target sites for the Flp recombinase: FRT3 can recombine with another FRT3, and FRT with FRT, but FRT3 cannot recombine with FRT. When recombined in cis, these mutually incompatible FRT target sites randomly excise one of two inserts (see *Bosch et al., 2015*). Hence, Flp expression will result in the formation of either T2A-$_{FRT3}$-LexA-GAD-$_{FRT}$ (caused by recombination between FRT3 sites) or T2A-$_{FRT3}$-QF2-$_{FRT}$ (if recombination occurs between FRT sites). Both recombination products

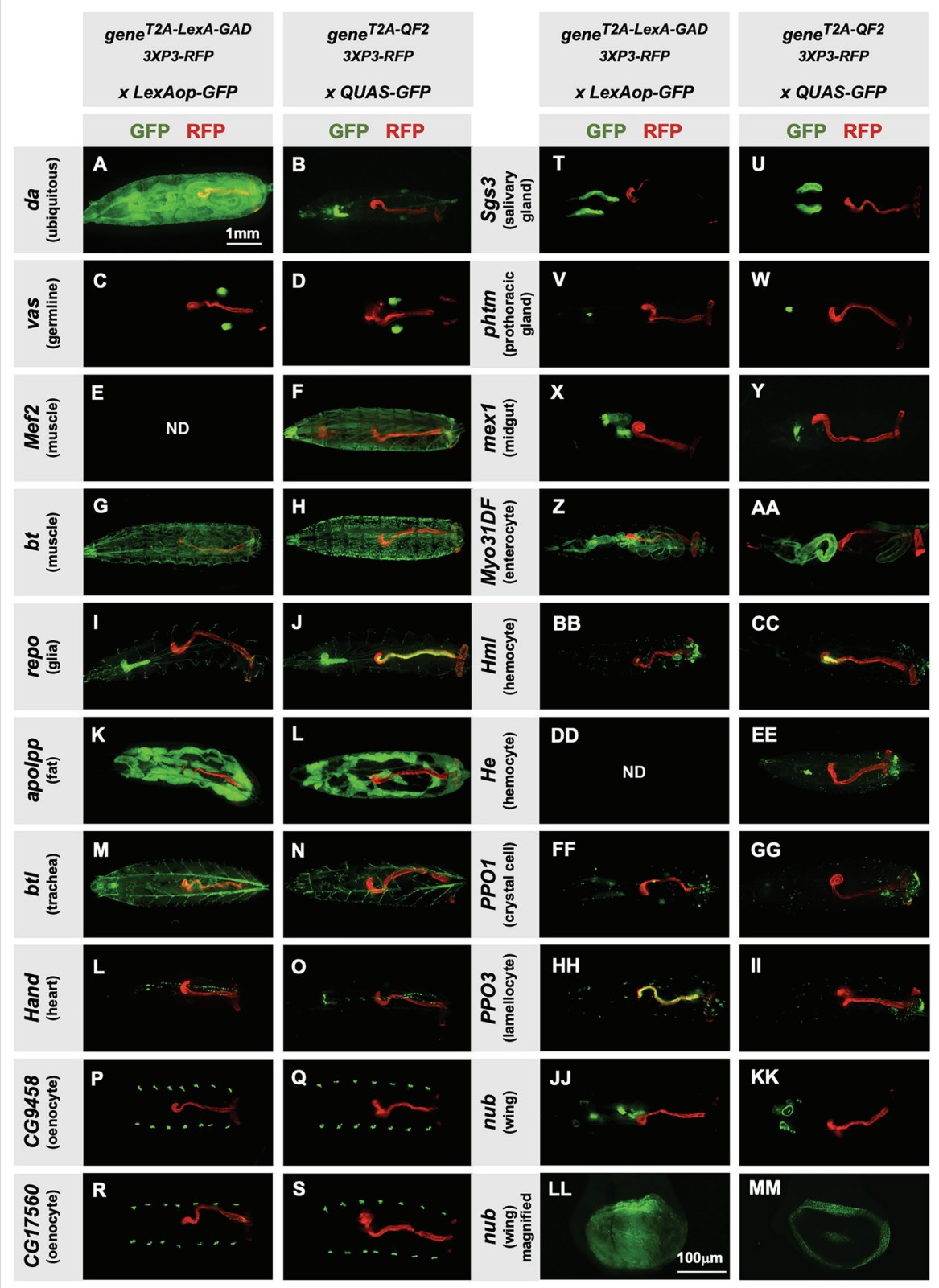

**Figure 2.** Tissue specificity of T2A-LexA-GAD and T2A-QF2 knock-ins. (**A–KK**) T2A-LexA-GAD knock-in lines crossed to a LexAop-GFP reporter and T2A-QF2 knock-in lines crossed to a QUAS-GFP reporter. Panels show third instar larva. GFP shows the driver line expression pattern. RFP shows the 3XP3 transformation marker, which labels the posterior gut and anal pads of the larva. Gene names and tissues are on the left. We failed to obtain LexA-GAD knock-ins for *Mef2* (**E**) and *He* (DD). (LL–MM) Third instar imaginal disc from the insertions in the *nubbin (nub)* gene. Note that most of the

*Figure 2 continued on next page*

*Figure 2 continued*

lines are highly tissue-specific and are comparable between the LexA-GAD and QF2 knock-ins. Insertions in the *daughterless* gene *(da) and nub* are an exception, as the T2A-LexA-GAD, but not the T2A-QF2, gives the expected expression pattern. Insertions in the gut-specific genes *mex1* (**X–Y**) and *Myo31Df* (**Z–AA**) also differed between the LexA-GAD and QF2 drivers.

will be stable even in the presence of Flp because they now lack a pair of compatible FRT sites. This enables us to obtain the individual driver lines (QF2 and LexA-GAD) by crossing the double drivers to flies that express Flp in the germline upon heat shock. We tested this with the *hh* and *dpp* lines and observed robust generation of both T2A-QF2 and T2A-LexA-GAD from *hs-Flp; T2A-QF2-T2A-LexA-GAD* parents (*Figure 5B*). In the case of the *hh* line, 15 out of 36 heat-shocked parents gave rise to at least one T2A-LexA-GAD progeny, with a mean of 14% recombinant offspring per parent. 20 out of 36 gave rise to at least one T2A-QF2 progeny, with a mean of 9% recombinant offspring per parent. In the case of the *dpp* line, 31 out of 32 heat-shocked parents gave rise to at least one T2A-LexA-GAD progeny, with a mean of 30% recombinant offspring per parent. 17 out of 32 gave rise to at least one T2A-QF2 progeny, with a mean of 9% recombinant offspring per parent. We verified the recombinants by crossing to a stock containing both LexAop-mCherry and QUAS-GFP. *Figure 5C–E* shows the wing disc from the *hh* double driver (*Figure 5C*), a recombinant which only expresses T2A-QF2 (*Figure 5D*) and a recombinant which only expresses LexA-GAD (*Figure 5E*). Recombinants were also independently verified by PCR of the insertions (*Figure 5F and G*), where we observed the expected smaller band sizes in the derivative T2A-QF2 and T2A-LexA-GAD relative to the parental double driver.

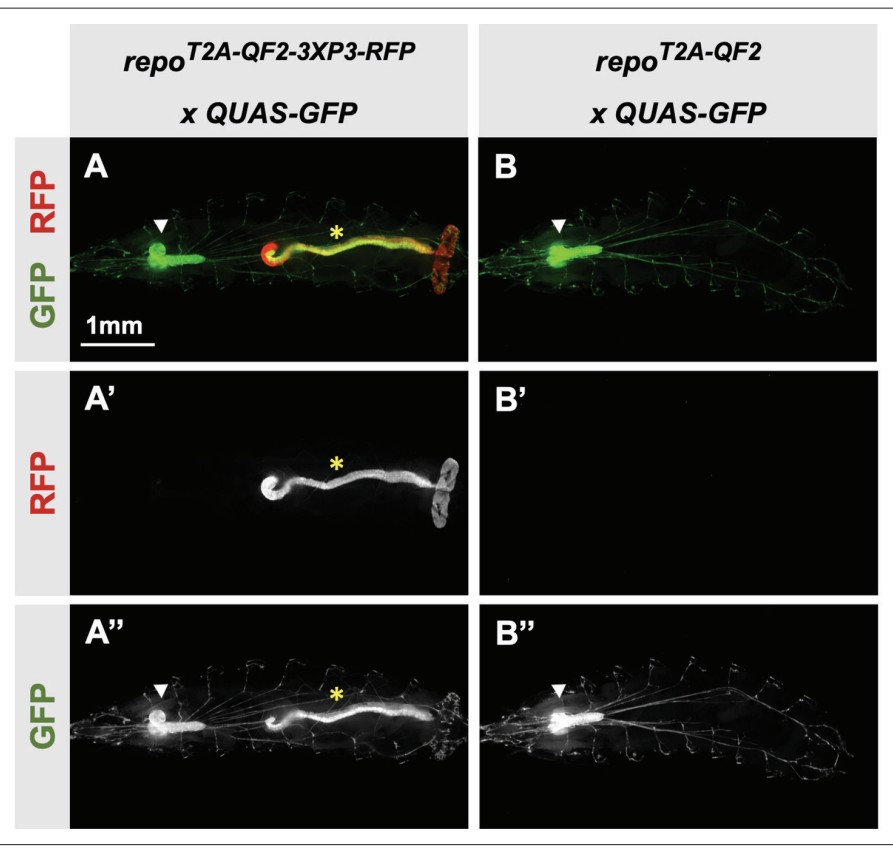

**Figure 3.** 3XP3-RFP can cause misexpression of T2A-LexA-GAD or T2A-QF2. (**A**) T2A-QF2-3XP3-RFP in the *repo* gene crossed to a QUAS-GFP reporter. In third instar larva, the reporter is expressed in the expected glial cells, but also misexpressed in gut and anal pad (yellow asterisk). (**B**) T2A-QF2 in the *repo* gene with the 3XP3-RFP removed by Cre-Lox recombination, crossed to a QUAS-GFP reporter. Removal of 3XP3-RFP eliminated gut and anal pad misexpression and did not affect glial cell expression (white arrowheads).

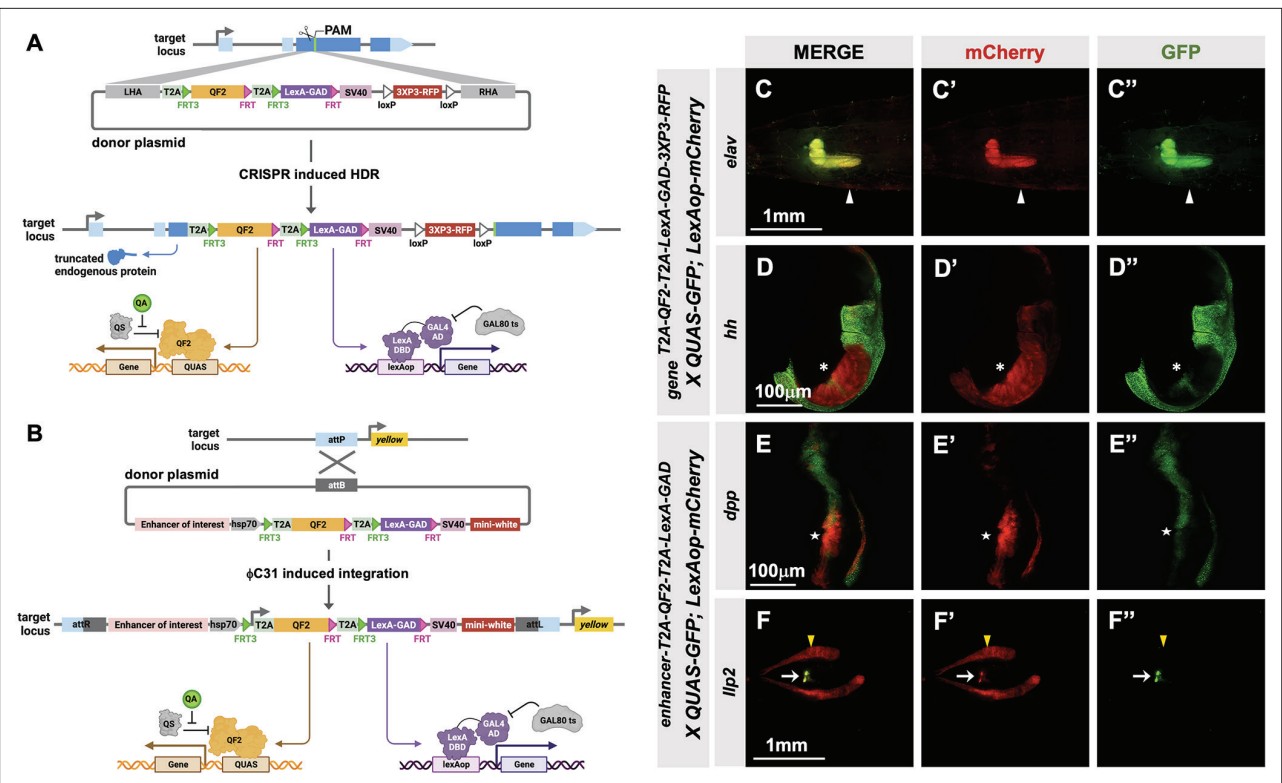

**Figure 4.** T2A-QF2-T2A-LexA-GAD double driver lines. (**A**) CRISPR-based HDR strategy for integration of the T2A-QF2-T2A-LexA-GAD-3XP3 in the most 5' coding exon common to all or most isoforms, resulting in expression of both activators under control of the endogenous gene regulatory region. The knock-in also produces a truncated endogenous protein and thus a strong loss-of-function allele. If desired, one of the two coding regions can then be excised with Flp, resulting in flies that express only QF2 or LexA-GAD. (**B**) Alternative strategy allows gene enhancers to be cloned upstream of of T2A-QF2-T2A-LexA-GAD. The vector backbone includes an *attB* site for phiC31 insertion into *attP* flies. (**C–D**) *T2A-QF2-T2A-LexA-GAD* knock-ins crossed to a QUAS-GFP+LexAop-mCherry double reporter line. (**C**) The *elav^T2A-QF2-T2A-LexA-GAD* knock-in drives both QUAS-GFP and LexAop-mCherry in the larval brain. There is some leakiness of mCherry in the body wall muscle (arrowheads). (**D**) The *hh^T2A-QF2-T2A-LexA-GAD* knock-in drives both QUAS-GFP and LexAop-mCherry in the posterior of the wing imaginal disc. GFP expression is much less than mCherry in the wing pouch (asterisks). (**E–F**) Enhancer-T2A-QF2-T2A-LexA-GAD lines crossed to a QUAS-GFP+LexAop-mCherry double reporter line. (**E**) The *dpp-blk* enhancer-T2A-QF2-T2A-LexA-GAD line drives both QUAS-GFP and LexAop-mCherry along the anterior/posterior boundary of the wing imaginal disc. GFP expression is much less than mCherry in the wing pouch (stars). (**F**) The *Ilp2* enhancer-T2A-QF2-T2A-LexA-GAD line drives both QUAS-GFP and LexAop-mCherry in the insulin-producing cells of the larval brain (arrows). The fat body mCherry expression (yellow arrowhead) is from leakiness of the reporter stock and does not indicate LexA-GAD activity.

## TRiP LexAop and QUAS shRNA vectors produce effective gene knockdown

The TRiP has previously generated a QUAS version of our standard shRNA expression vector, pQUAS-WALIUM20 (*Perkins et al., 2015*), containing the standard five copies of the QF-binding site (*Potter et al., 2010*). We also generated pLexAop-WALIUM20, containing 13 LexA DNA-binding sites, previously reported to give optimal expression with minimal leakiness (*Pfeiffer et al., 2010*; *Figure 6A*). We cloned shRNAs targeting *forked (f)* and *ebony (e)* genes into these vectors and assayed their phenotypes when crossed to ubiquitous LexA-GAD and QF2 drivers. The first driver tested was the T2A-LexA-GAD knock-in in the *da* gene (*Figure 2A*), generated in this study. This produced 100% penetrant forked bristle and ebony cuticle phenotypes in the adult thorax when crossed to *f* (*Figure 6E*) and *e* (*Figure 6H*) shRNA lines, respectively. For this experiment, *white (w)* shRNA was used as a negative control. We did not test the T2A-QF2 knock-in in the *da* gene, as we have already shown that the expression pattern was not ubiquitous (*Figure 2G*). Instead, to directly compare the two systems, we used previously described ubiquitous LexA-GAD and ubiquitous QF2 (*Lai and Lee, 2006*) under the control of αTub84B regulatory sequences. Both Tub-LexA-GAD and Tub-QF2 drivers generated knockdown phenotypes in the thorax when crossed to *f* and *e* shRNA lines. However,

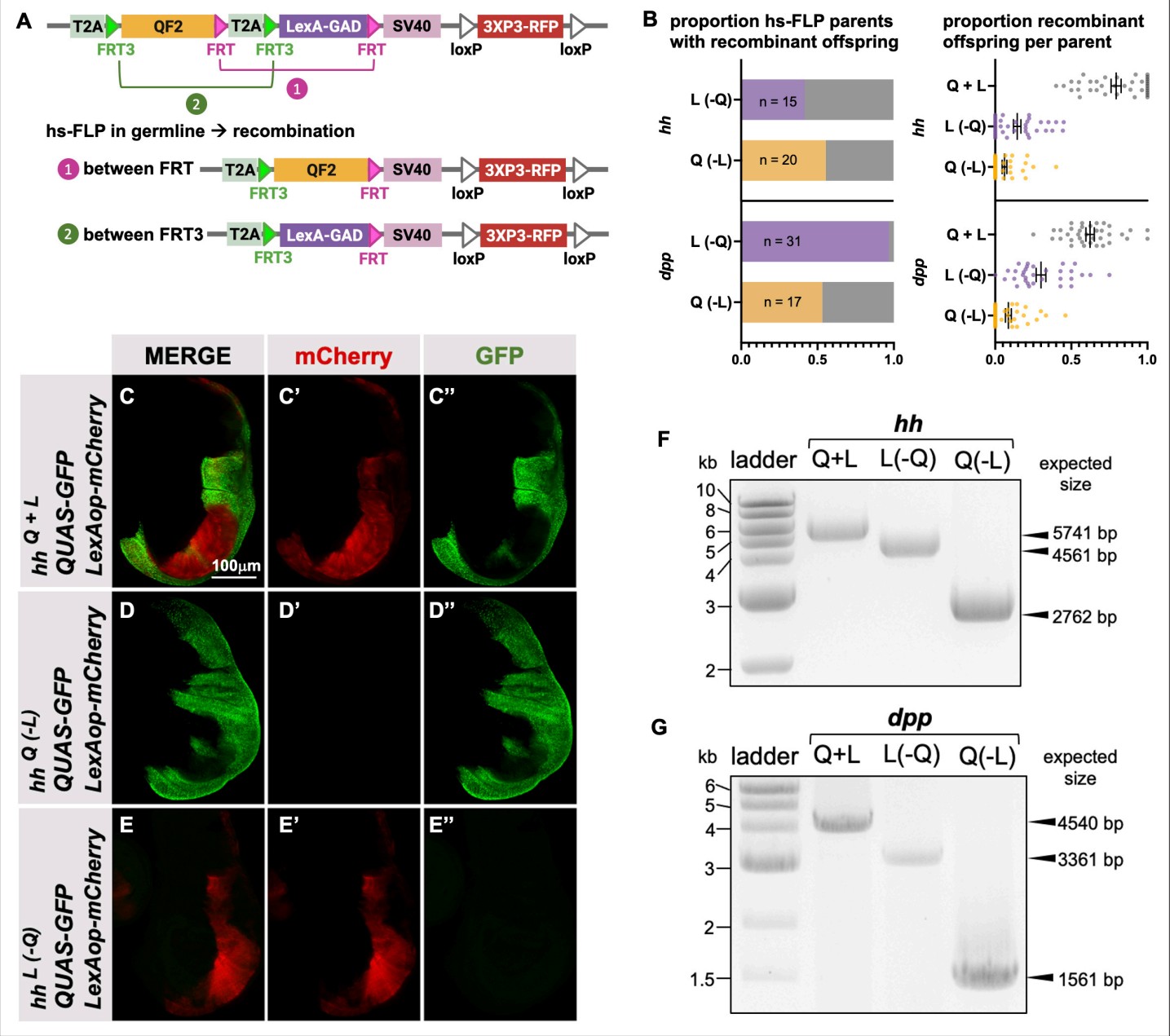

**Figure 5.** Generation of single drivers from T2A-QF2-T2A-LexA-GAD knock-ins by hs-FLP. (**A**) FLP/FRT recombination scheme. Flies containing both hs-FLP and a T2A-QF2-T2A-LexA-GAD knock-in are heat shocked during larval development to induce one of two mutually exclusive recombination events in their germline between either FRT or FRT3. (**B**) Heat shock of *hsFLP; hh^T2A-QF2-T2A-LexA-GAD* and hsFLP; *dpp^T2A-QF2-T2A-LexA-GAD* flies produces frequent recombinants, both T2A-QF2 and T2A-LexA-GAD. The bar graph shows the proportion of heat-shocked animals that produced at least one recombinant offspring. The dot plot shows the proportion of recombinant offspring per heat-shocked parent. Mean ± SD is indicated. (**C–D**) Validation of individual *hh^T2A-QF2* and *hh^T2A-LexA-GAD* derivatives by immunofluorescence. All panels show third instar larval wing discs dissected from potential *hh^T2A-QF2-T2A-LexA-GAD* recombinants crossed to a QUAS-GFP+LexAop-mCherry reporter line. (**C**) Wing disc from non-recombinant *hh^T2A-QF2-T2A-LexA-GAD* showing expression of both GFP and mCherry in the posterior of the wing disc. Note, this is the same image as shown in *Figure 4D*. (**D**) Wing disc from recombinant *hh^T2A-QF2* showing expression of GFP but not mCherry in the posterior of the wing disc. (**E**) Wing disc from recombinant *hh^T2A-LexA-GAD* showing expression of mCherry but not GFP in the posterior of the wing disc. Validation of (**F**) *hh^T2A-QF2* and *hh^T2A-LexA-GAD* derivatives and (**G**) *dpp^T2A-QF2* and *dpp^T2A-LexA-GAD* derivatives by PCR from genomic DNA from individual flies. In all panels, for brevity, T2A-QF2-T2A-LexAop, T2A-QF2, and T2A-LexAop, are notated as Q+L, Q(-L), and L(-Q), respectively.

The online version of this article includes the following source data for figure 5:

**Source data 1.** Original file for the DNA gel analysis in *Figure 5F and G*.

**Source data 2.** Original file for the DNA gel analysis with boxes indicating the portions cropped to produce the image for *Figure 5F and G*.

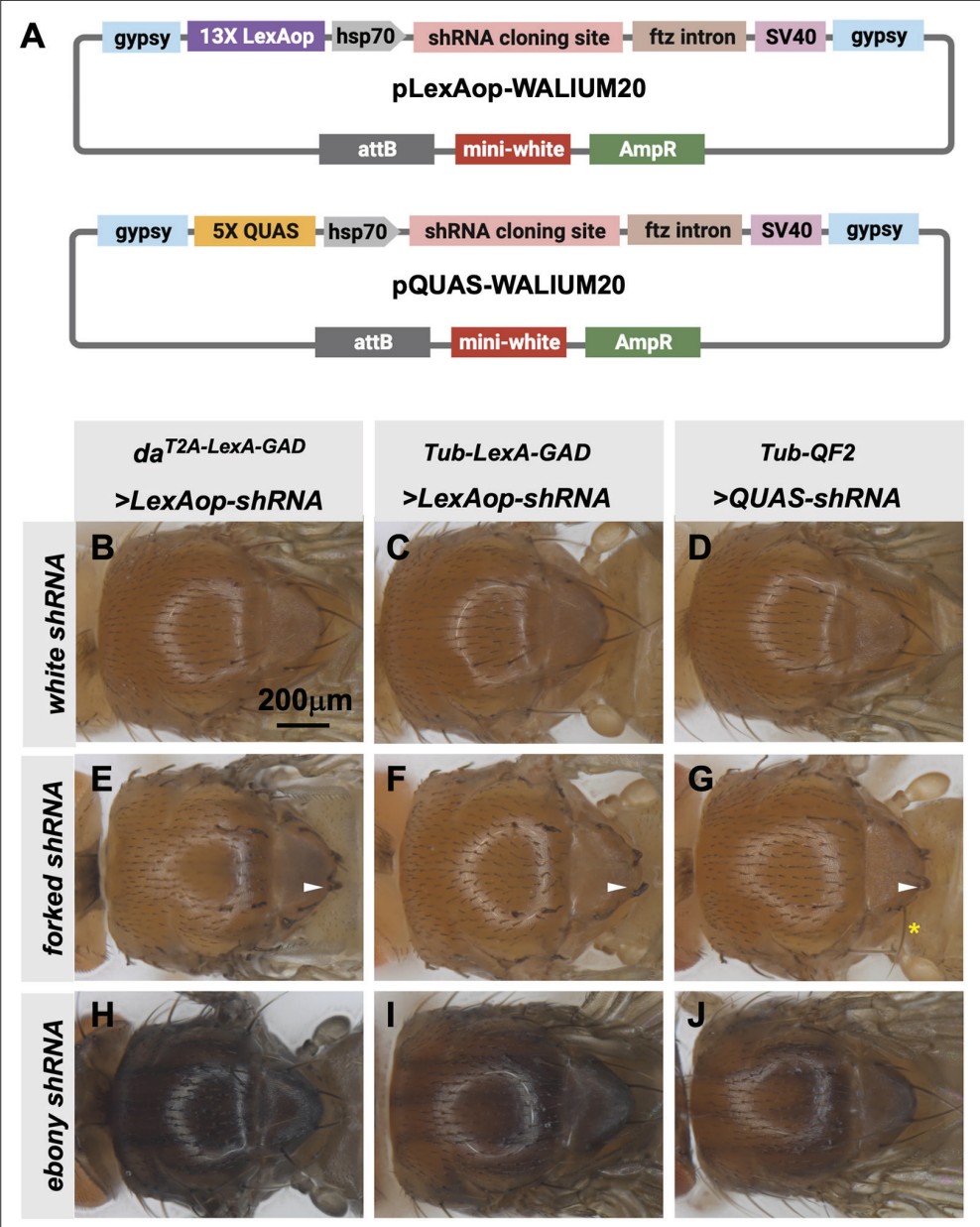

**Figure 6.** Transgenic RNAi Project (TRiP) LexAop and QUAS shRNA vectors produce effective gene knockdown. (**A**) shRNAs for knockdown or genes for overexpression were cloned into pLexAop-WALIUM20 and pQUAS-WALIUM20, derived from the TRiP WALIUM20 vector. (**B–J**) Dorsal view of adult fly thoraces resulting from crosses of LexAop or QUAS shRNAs to *da^T2A-LexA-GAD*^ (generated in this study), *Tub-LexA-GAD* (Bloomington *Drosophila* Stock Center [BDSC] 66686), or *Tub-QF2* (BDSC 51958). (**B–C**) *white* shRNA control produced no thoracic phenotypes in any of the crosses. (**E–G**) *forked* shRNA produced a forked bristles phenotype (white arrowheads). Note that some bristles retain a more elongated wild-type morphology with the Tub-QF2-driven

*Figure 6 continued on next page*

*Figure 6 continued*

*forked* knockdown (G, yellow asterisk). (**H–J**) *ebony* shRNA produced a darkened cuticle phenotype. The *da*$^{T2A\text{-}LexA\text{-}GAD}$ driver produced the strongest phenotype (compare panel H to I and J).

the Tub-LexA-GAD phenotypes were stronger than those of Tub-QF2 (*Figure 6C–D, F–G, I–J*). For example, Tub-LexA-GAD produced a fully penetrant *f* bristle phenotype (*Figure 6F*) while some wild-type bristles remained on the thoraces of Tub-QF2 *f* knockdown (*Figure 6G*). Neither Tub-LexA-GAD nor Tub-QF2 was able to achieve the strength of phenotype generated by the T2A-LexA-GAD *da* knock-in line (compare the darkness of the cuticle caused by *e* knockdown in *Figure 6H–J*).

## Discussion

There have been previous efforts to make LexA and QF tools, but the availability of LexA-GAD and QF2 tissue-specific fly stocks remains a major resource gap, preventing the average fly researcher from performing the type of multi-tissue manipulations that are essential to advance the study of organ and tissue communication. The collection of T2A-LexA-GAD and T2A-QF2 drivers described here is unique in that it is the first to focus on covering the major organ systems of the fly. Our protocol for generating these stocks is straightforward and can be easily adapted to produce driver lines for other fly tissues. These efforts will benefit from the emergence of scRNAseq datasets, which can be used to identify target genes with highly organ or tissue-specific gene expression patterns. For example, we identified the uncharacterized genes *CG9458* and *CG17560,* as highly oenocyte-specific based on the Fly Cell Atlas Single-cell transcriptome (*Li et al., 2022*), and the T2A-LexA-GAD and T2A-QF2 knock-ins in these genes were indeed restricted to this tissue. Our collection of T2A-LexA-GAD and T2A-QF2 and double driver vectors can be easily adapted to target any gene for CRISPR knock-in, with a high probability that the resulting line will accurately reflect the expression of the endogenous locus. The specificity achieved with this approach can also be seen in recent efforts to build collections of gene-specific T2A-Split-GAL4 and T2A-GAL4 insertions (*Kanca et al., 2019*; *Chen et al., 2023*; *Ewen-Campen et al., 2023*). Our vectors are compatible with both the traditional large HA flanked cassettes and the more streamlined drop-in approach. In our hands, the drop-in cloning strategy is particularly effective, as the cloning success rate is 100%, requires little troubleshooting, has a very high knock-in rate, and only costs ~$100 to synthesize the construct (*Kanca et al., 2019*; *Kanca et al., 2022*).

Our results also present the opportunity to directly compare the LexA-GAD and QF2 systems. While we had no difficulty obtaining knock-ins for both types of activators, we did observe that for some target genes, the T2A-QF2 was only active in a subset of the expected gene expression pattern. In particular, we found that T2A-QF2 was difficult to express in the wing pouch. Additionally, we found that the driver expression in the gut-specific genes, *mex1* and *Myo31Df,* differed between the LexA-GAD and QF2 transformants. In both cases the LexA-GAD was more broadly expressed along the length of the gut than the QF2. It may be that toxicity is an issue, and the weaker QF2w may be a better option for generating drivers in some organs (*Riabinina and Potter, 2016*). Alternatively, differences in the LexA-GAD and QF2 sequences, and sequence length, could impact the function of nearby gene regulatory regions. Further, we expect that there will also be differences between the expression pattern of corresponding GAL4 and the LexA-GAD/QF lines, as the latter were made by knock-in, while the former are often enhancer traps. However, based on our larval mounts and dissections, the stocks generated in this paper are highly specific to the expression pattern of the targeted genes. Importantly, our knock-in constructs contain the 3XP3-RFP cassette for screening transformants. Perhaps due to interaction between the 3XP3 promoter and the regulatory regions of the target gene, we occasionally saw misexpression of the LexA-GAD/QF2 in the 3XP3 domain. We have therefore prioritized Cre-Lox removal of the 3XP3-RFP cassette from our knock-in stocks, and advise that users of the plasmids described here likewise remove the marker, following successful knock-in.

When we compared the knockdown efficiency of shRNAs targeting *forked* and *ebony* and, we found that the TRiP 13XLexAop vector was more effective than the 5XQUAS vector, although both were able to induce knockdown. Based on these results, the TRiP is currently generating a set of ~100 LexAop shRNA lines encompassing the genes targeted by the most commonly ordered UAS shRNA stocks at the BDSC. See *Supplementary file 1* for a list of all LexAop shRNA lines in production. There

remains an unmet need for a single vector that would allow for UAS/LexAop/QUAS control of different shRNAs. However, recent innovations in multi-module vectors and multiplexed drug-based genetics allow researchers to more efficiently generate UAS/QUAS/lexAop transgenic fly strains (*Matinyan et al., 2021*; *Wendler et al., 2022*). In summary, we have generated a set of stocks, vectors, and protocols that when combined with the wide array of GAL4 lines will greatly expand the ability of *Drosophila* researchers to modulate gene expression in multiple tissues simultaneously.

## Materials and methods

### Generation of pHDR-T2A-LexA/QF2-Hsp70-3xP3-RFP plasmids

To generate the pHDR-T2A-LexA/QF2-Hsp70-3xP3-RFP plasmids, we replaced the SV40-3'UTR present in the pCRISPaint-T2A-LexA/QF2 vector (*Bosch et al., 2020*) with the hsp70-3'UTR, using Gibson assembly (*Gibson et al., 2009*) (NEB E2611). The hsp70-3'UTR was amplified from a pCRISPaint-GAL4-Hsp70 plasmid using the following primers: F: GTCGACTAAAGCCAAATAG, R: AAACGAGTTTTTAAGCAAAC, appended at the 5' end with appropriate homologous overhangs for Gibson assembly. To remove the two endogenous SacI sites in the QF2 coding sequence, we used Gibson assembly featuring primers that introduce synonymous SNPs which mutate the SacI-binding sites without disrupting the coding sequence (GAGCTC>GAACTC).

### Cloning of T2A-LexA-GAD and T2A-QF2 donor constructs

For the long-HA cloning method, we amplified the HAs by selecting ~1000 kb upstream and downstream of the guide cut site, making sure that the left HA is in frame with the T2A, and that the ends of the primers contain Gibson overhangs matching the pHDR-T2A-LexA/QF2-Hsp70-3xP3-RFP plasmids. Amplification was always from genomic DNA from the nos-Cas9 injection stock. We used Phusion (NEB), Taq (Takara), or Q5 (NEB). In cases where the PCR product was faint, we set up eight PCR samples in parallel, combined them, concentrated them using phenol-chloroform extraction followed by ethanol precipitation, and ran the concentrated sample on a gel to obtain a bright band that was then gel-purified for use in Gibson assembly. Once the HAs were amplified, we performed Gibson assembly with the pHDR-T2A-LexA/QF2-Hsp70-3xP3-RFP plasmids digested with AscI/SacI. Guide RNAs were cloned separately in pCFD3 (*Port et al., 2014*). We chose previously designed gRNAs from https://www.flyrnai.org/crispr3/web/. Our criteria were: efficiency >5, and no U6 termination site. Primer and guide sequences are in *Supplementary file 2*. We designed sense and antisense oligos for each gRNA, and then annealed them together to make a ds-oligo with overhangs for cloning: we combined 1.0 µl each 100 µM sense+antisense oligo, 1.0 µl 10× T4 ligase buffer, 0.5 µl T4 polynucleotide kinase (NEB), and 6.5 µl dH$_2$O, and incubated at 37°C 30 min followed by 5 min at 95°C and slowly cooling down to room temperature (–5°C/min). The ds-oligos were then ligated into BbsI-digested pCFD3 vector with T4 ligase (NEB). Following cloning, plasmids were verified by sequencing with primer (GCCGAGCACAATTGTCTAGAATGC).

For the drop-in method, we followed a modified version of the protocol described elsewhere (*Kanca et al., 2022*). Briefly, homology donor intermediate vectors were ordered from Genewiz in the pUC57 Kan_gw_OK2 vector backbone, containing the gene-specific guide sequence, 200 bp short HAs flanking the genomic cut site, and a BbsI and SacI cloning site. pHDR-T2A-LexA/QF2-Hsp70-3xP3-RFP plasmids were digested with AscI/SacI, producing a 2677 bp fragment for QF2 and the ~4.5 kb fragment for LexA-GAD, each with overhangs compatible with the pUC57 Kan_gw_OK2 BbsI/SacI overhangs. The digested pUC57 Kan_gw_OK2 backbone, containing the HAs and guides, was then ligated with the digested T2A-LexA/QF2-Hsp70-3xP3-RFP with 2.5 µl 10× T4 DNA ligase buffer (NEB B0202S) and 0.5 µl T4 DNA ligase (NEB M0202S). Sequences of the synthesized drop-in fragments are in *Supplementary file 3*.

### Construction of T2A-QF2-T2A-LexA-GAD double driver constructs

For combined expression of LexA-GAD and QF2 transcription factors, we built two different vectors: (1) a CRISPR donor version, which we used to insert the LexA-GAD-QF2 cassette into an endogenous locus of interest, such that the expression of lexA-GAD and QF2 is driven by endogenous regulatory sequences, and (2) a φC31-attB version, which is used to clone an enhancer fragment of interest and integrated into an attP site in the fly genome.

To build the pHDR-T2A-QF2-T2A-LexA-GAD-3XP3-RFP construct, we used pCRISPaint-T2A-QF2 and pCRISPaint-T2A-LexA-GAD vectors (**Bosch et al., 2020**) to assemble the the vector as follows (see also **Supplementary file 4**):

1. A gBlock double-stranded DNA fragment covering the N-terminus of the QF2 ORF along with T2A, FRT3, and HAs for Gibson assembly (BJusiak-QF2-N).
2. Part of the QF2 ORF amplified as a PCR product off the CRISPaint-T2A-QF2 vector using BJusiak-QF2-fwd+rev primer pair.
3. A gBlock encoding the C-terminus of QF2 and the FRTwt-T2A-FRT3 sequence between QF2 and LexA-GAD, along with HAs for Gibson assembly (BJusiak-QF2-C).
4. Most of the LexA-GAD ORF, PCR-amplified off the CRISPaint-T2A-LexA-GAD template with the BJusiak-lexA1-fwd+rev primer pair.
5. BJusiak-FRToligo1-top+bottom, a pair of single-stranded oligos annealed to make a ds-oligo encoding the FRTwt site 3' of LexA-GAD.
6. CRISPaint vector digested with ApaI+KpnI restriction enzymes.

gBlocks and oligos were ordered from Integrated DNA Technologies (IDT) and restriction enzymes were from NEB. The sequences of all gBlocks and oligos used to build pHDR-QF2-LexA-GAD are in Supplementary materials and methods. HAs and guides for CRISPR were cloned as described above for the T2A-LexA-GAD and T2A-QF2 single donor constructs.

We digested CRISPaint-T2A-QF2 with ApaI at 25°C in CutSmart buffer, followed by digestion with KpnI-HF at 37°C. We ran the digest on an agarose gel and purified the 4.9 kb vector backbone. We then set up the Gibson assembly:

> Vector backbone (4,938 bp) 100 ng
> lexGAD PCR product (2875 bp) 117 ng 2:1 insert:vector molar ratio
> QF2 PCR product (292 bp) 18 ng 3:1 insert:vector
> BJusiak-QF2-N (713 bp) 44 ng 3:1 insert:vector
> BJusiak-QF2-C (324 bp) 20 ng 3:1 insert:vector
> FRT-ds-oligo (88 bp) 9.0 ng 5:1 insert:vector.

We added $dH_2O$ to 10.0 µl final volume, added 10.0 µl Gibson Assembly Mix (NEB), and incubated at 50°C for 1 hr. We transformed 5.0 µl of the Gibson assembly reaction into Top10 chemically competent *Escherichia coli* and plated on LB+carbenicillin, then screened *E. coli* colonies by PCR with the BJusiak-Qlex-test1-fwd+rev primer pair (**Supplementary file 4**), expected to give a 0.7 kb product in the presence of correctly assembled CRISPaint-QF2-lexGAD. Plasmid DNA was prepared from positive colonies using the ZymoPure midiprep kit. Restriction digest fingerprinting of the plasmid midipreps produced expected band patterns, which were verified by sequencing.

To build the pMCS-T2A-QF2-T2A-LexA-GAD-WALIUM20 construct, we performed Gibson assembly with the following fragments (see also **Supplementary file 4**):

1. MCS-WAL20-START gBlock, including the MCS, hsp70 promoter, T2A-FRT3 coding sequence, and HAs for Gibson assembly;
2. MCS-WAL20-STOP-v2 gBlock, including the 3' end of LexA-GAD ORF, FRT coding sequence, and STOP codon;
3. QF2-T2A-LexA-GAD ORF PCR-amplified with FRT-QF2-fwd+FRT-lexA-GAD-rev primer pair – used Hot-Start Q5 polymerase with GC enhancer (NEB);
4. WALIUM20 vector digested with BamHI+EcoRI.

We transformed the Gibson assembly reaction into Top10 chemically competent *E. coli* and plated them on LB+carbenicillin. We screened colonies with PCR using the BJusiak-Qlex-test1-fwd+rev primer pair, same as for CRISPaint-QF2-lexGAD. Positive colonies were used for plasmid preps, which were sent for Sanger sequencing (Azenta) with the QF2-seq1-rev primer (TGTTAGTGAGATCAGC GAAC expected to read across the MCS-hsp70 region).

To make a variant pMCS-QF2-LexA-GAD-alt that lacks the Hsp70 promoter, we did Gibson assembly as above, except we replaced the MCS-WAL20-START with MCS-WAL20-START-new gBlock, which lacks the Hsp70 sequence.

## Cloning HAs into pHDR-T2A-QF2-T2A-LexA-GAD-3XP3-RFP

HAs were amplified by PCR as described above. See also **Supplementary file 5** for the PCR primers. We then digested pHDR-T2A-QF2-T2A-LexA-GAD-3XP3-RFP separately with AscI+SaCI to release

the vector backbone and with NotI+SacI to purify the T2A-QF2-T2A-lexA-GAD-3XP3-RFP fragment. We gel-purified the backbone, T2A-QF2-T2A-lexA-GAD-3XP3-RFP fragment, and the HA PCR products, and we assembled all four fragments using Gibson assembly. We used 50 ng vector backbone, twofold molar excess of T2A-QF2-T2A-lexA-GAD-3XP3-RFP, and threefold molar excess of each HA. We transformed the Gibson assembly product into Top10 chemically competent *E. coli*, miniprepped (QIAGEN) and verified by sequencing. Guides were cloned into pCFD3 as described above.

## Cloning large enhancer fragments into pMCS-T2A-QF2-T2A-lexA-GAD-WALIUM20

*Ilp2*-GAL4 has been described (*Wu et al., 2005*). The *dpp-blk* enhancer was described as a '4 kb BamHI fragment' (*Masucci et al., 1990*) that is 17 kb 3′ of the *dpp* transcribed region (*Blackman et al., 1987*; *Blackman et al., 1987*). The primers used to PCR these fragments, using fly genomic DNA as template, are in *Supplementary file 6*. After PCR-amplifying the enhancer fragment with Q5 polymerase (+GC for *Ilp2*, no GC for *dpp*), we digested it and the destination vector with the corresponding enzymes (NotI+EcoRI for *dpp-blk*, PacI+EcoRI for *Ilp2*). We used pMCS-T2A-QF2-T2A-lexA0GAD-WALIUM20 for dpp-blk and pMCS-T2A-QF2-T2A-lexGAD-WALIUM20-alt (which lacks the hsp70 promoter) for *Ilp2*, since *dpp-blk* does not have a basal promoter, but the *Ilp2* enhancer does. We ligated the PCR fragments into the vectors using T4 ligase (NEB), transformed into *E. coli*, miniprepped (QIAGEN) and verified by sequencing.

All vectors described here that are required to produce new driver lines will be made available at Addgene.

## Cloning shRNAs

shRNAs (21 bp) were cloned into pQUAS-WALIUM20 and pLexAop-WALIUM20 vectors digested with EcoRI+XbaI, as described previously (*Ni et al., 2011*). The oligos were as follows:

| Gene | Oligo forward | Oligo reverse |
|---|---|---|
| white | ctagcagtCAGCGTCGTCCAGGTGCTGAA agtttatattcaagcataTTCAGCACCTGGACGACGC TGgcg | aattcgcCAGCGTCGTCCAGGTGCTGAA tagttatattcaagcataTTCAGCACCTGGACGACGCTGa ctg |
| ebony | ctagcagtTCCGGAGAGGTTCTTGGAGAA tagttatattcaagcataTTCTCCAAGAACCTCTCCGGAg cg | aattcgcTCCGGAGAGGTTCTTGGAGAA tagttatattcaagcataTTCTCCAAGAACCTCTCCGGAa ctg |
| forked | ctagcagtTCCGACCTAATTGCCGAGCTA tagttatattcaagcataTAGCTCGGCAATTAGGTCGGAg cg | aattcgcTCCGACCTAATTGCCGAGCTA tagttatattcaagcataTAGCTCGGCAATTAGGTCGGAa ctg |

All transgenic lines were sequenced to confirm the identity of the shRNA.

## Fly injections

All CRISPR constructs were injected at 250 ng/μl along with 100 ng/μl gene-specific gRNA(s) where appropriate. 300 embryos from *y w; iso18; attP2, nos-Cas9* for genes on the X, second or fourth chromosomes and *y w; attP40, nos-Cas9; iso5* for genes on the third chromosome per genotype were injected as described previously (*Lee et al., 2018*). Knock-in efficiencies were comparable to previous reports (*Kanca et al., 2019*; *Kanca et al., 2022*). Resulting G0 males and females were crossed individually to appropriate balancer flies for *3XP3-RFP* screening. Positive lines were balanced, and stocks were established. For phiC31-integration, each plasmid was injected at 50 ng/μl into *y v nos-phiC31-int; attP40* (for chromosome 2 insertions) or *y v nos-phiC31-int; attP2* (for chromosome 3 insertions). Injected male G0 flies were crossed with *y w; Gla/CyO* or *y w; Dr e/TM3, Sb* to identify transformants and remove the integrase from the X chromosome, and subsequently balanced.

## PCR validation of knock-ins

PCR primers that flank the integration site were designed for each targeted gene. These primers were used in combination with primers that bind within the inserted cassette in both orientations. 500–800 nt amplicons were amplified from genomic DNA from individual insertion lines through single fly PCR using GoTaq green master mix (Promega M7122).

All transgenic fly stocks described here will be made available at the BDSC.

## Imaging

T2A-LexA-Gad, T2A-QF2, or double driver lines were crossed to *y w; Sp/CyO; LexAop-GFP (BDSC 52266), y w; QUAS-GFP/CyO (BDSC 52264), or y w; QUAS-GFP (BDSC 52264)/CyO-GFP; LexAop-RFP* (BDSC 52271)/*TM3*. Larvae were placed in PBS and sandwiched between a slide and coverslip, then live-imaged using a Zeiss (Carl Zeiss, Thornwood, NY, USA) Stemi SVII fluorescence microscope. Wing imaginal discs from third instar larvae were dissected in PBS, fixed in 4% methanol-free formaldehyde, and permeabilized in PBT, mounted on glass slides with vectashield (H-1000; Vector Laboratories) under a coverslip, and imaged on a Zeiss 780 confocal microscope.

## Acknowledgements

We thank Stephanie Mohr for her support and advice on this project. This work was supported by 5P41GM132087 and 5R24OD030002. NP is an HHMI investigator. JAB was supported by the Damon Runyon Foundation (DRG-2258-16) and a 'Training Grant in Genetics' T32 Ruth Kirschstein-National Research Service Award institutional research training grant funded through the NIH/National Institute of General Medical Sciences (T32GM007748).

## Additional information

### Funding

| Funder | Grant reference number | Author |
| --- | --- | --- |
| National Institute of General Medical Sciences | 5P41GM132087 | Jonathan Zirin |
| NIH Office of the Director | 5R24OD030002 | Jonathan Zirin |
| Damon Runyon Cancer Research Foundation | DRG-2258-16 | Justin A Bosch |
| National Institute of General Medical Sciences | T32GM007748 | Justin A Bosch |
| Howard Hughes Medical Institute | | Norbert Perrimon |

The funders had no role in study design, data collection and interpretation, or the decision to submit the work for publication.

### Author contributions

Jonathan Zirin, Conceptualization, Resources, Formal analysis, Supervision, Funding acquisition, Validation, Investigation, Visualization, Methodology, Writing – original draft, Project administration, Writing - review and editing; Barbara Jusiak, Conceptualization, Formal analysis, Investigation, Methodology, Writing – original draft; Raphael Lopes, Alexandria Risbeck, Corey Forman, Christians Villalta, Investigation; Benjamin Ewen-Campen, Investigation, Methodology, Writing – original draft; Justin A Bosch, Conceptualization, Methodology, Writing - review and editing; Yanhui Hu, Data curation, Software, Formal analysis; Norbert Perrimon, Conceptualization, Formal analysis, Supervision, Funding acquisition, Methodology, Writing – original draft, Project administration, Writing - review and editing

### Author ORCIDs

Jonathan Zirin http://orcid.org/0009-0003-8242-5044
Barbara Jusiak https://orcid.org/0000-0001-6439-0456
Justin A Bosch http://orcid.org/0000-0001-8499-1566
Norbert Perrimon http://orcid.org/0000-0001-7542-472X

Reviewer #1 (Public Review): https://doi.org/10.7554/eLife.94073.3.sa1
Reviewer #2 (Public Review): https://doi.org/10.7554/eLife.94073.3.sa2

Author response https://doi.org/10.7554/eLife.94073.3.sa3

## Additional files

### Supplementary files
• Supplementary file 1. Table listing the gene names and antisense sequence of 96 constructs cloned and injected to produce a set of transgenic LexAop-shRNA fly lines.

• Supplementary file 2. Table listing the gene names, guide sequences, and primers for amplification of long homology arms for T2A-QF2 and T2A-LexA-GAD constructs.

• Supplementary file 3. Sequences of DNA fragments synthesized into the pUC57 Kan_gw_OK2 vector backbone for subsequent drop-in cloning.

• Supplementary file 4. Sequences of DNA fragments used to build pHDR-T2A-QF2-T2A-LexA-GAD and pMCS-T2A-QF2-T2A-LexA-GAD-WALIUM20. FRT3 sequences are shown in blue and FRTwt sequences are shown in red.

• Supplementary file 5. Primers and guide sequences for amplification of long homology arms for pHDR-T2A-QF2-T2A-LexA-GAD-3XP3-RFP constructs.

• Supplementary file 6. Primers used to amplify large enhancers for cloning into pMCS-T2A-QF2-T2A-LexA-GAD-WALIUM20. The dpp-blk primers have NotI and EcoRI sites for cloning into the MCS, and the Dilp2-enh primers have PacI and EcoRI sites.

• MDAR checklist

### Data availability
The current manuscript did not generate any datasets. Raw gel image source files are provided for Figure 5.

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
