## [Editor Report · eLife assessment]

This **important** study reports the generation of genetic tools for manipulating several tissues at the same time in *Drosophila*. The authors provide **convincing** evidence that this allows for the generation of LexA and QF2 driver lines, which will be of great utility for understanding inter-organ communication. Making the tools available through the *Drosophila* stock center and plasmid depository will ensure that they are easily accessed by many researchers.

---

## [Referee Report · Reviewer #1 (Public Review)]

Summary:

"Expanding the *Drosophila* toolkit for dual control of gene expression" by Zirin et al. aims to develop resources for simultaneous independent manipulation of multiple genes in *Drosophila*. The authors use CRISPR knock-ins to establish a collection of T2A-LexA and T2A-QF2 transgenes with expression patterns in a number of commonly studied organs and tissues. In addition to the transgenic lines that are established, the authors describe a number of plasmids that can be used to generate additional transgenes, including a plasmid to generate a dual insert of LexA and QF that can be resolved into a single insert using FLP/FRT-mediated recombination, and plasmids to generate RNAi reagents for the LexA and QF systems. Finally, the authors demonstrate that a subset of the LexA and QF lines that they generated can induce RNAi phenotypes when paired with LexAop or QUAS shRNA lines. In general, the claims of the paper are well supported by the evidence and the authors do a thorough job of validating the transgenic lines and characterizing their expression patterns.

---

## [Referee Report · Reviewer #2 (Public Review)]

Zirin, Jusiak, and Lopes et al presented an efficient pipeline for making LexA-GAD and QF2 drivers. The tools can be combined with a large collection of existing GAL4 drivers for a dual genetic control of two cell populations. This is essential when studying inter-organ communications since most of the current genetic drivers are biased toward the expression of the central nervous system. In this manuscript, the authors described the methodology for efficiently generating T2A-LexA-GAD and T2A-QF2 knock-ins by CRISPR, targeting a number of genes with known tissue-specific expression patterns. The authors then validated and compared the expression of double as well as single drivers and found the tissue-specific expression results were largely consistent as expected. Finally, a collection of plasmids for LexA-GAD and QF,2 as well as the corresponding LexAop and QUAS plasmids were generated to facilitate the expansion of these tool kits. In general, this study will be of considerable interest to the fly community and the resources can be readily generalized to make drivers for other genes. I believe this toolkit will have a significant, immediate impact on the fly community.

---

## [Author Response]

The following is the authors’ response to the original reviews.

**Public Reviews:**

**Reviewer #1 (Public Review):**
Summary:"Expanding the *Drosophila* toolkit for dual control of gene expression" by Zirin et al. aims to develop resources for simultaneous independent manipulation of multiple genes in *Drosophila*. The authors use CRISPR knock-ins to establish a collection of T2A-LexA and T2A-QF2 transgenes with expression patterns in a number of commonly studied organs and tissues. In addition to the transgenic lines that are established, the authors describe a number of plasmids that can be used to generate additional transgenes, including a plasmid to generate a dual insert of LexA and QF that can be resolved into a single insert using FLP/FRT-mediated recombination, and plasmids to generate RNAi reagents for the LexA and QF systems. Finally, the authors demonstrate that a subset of the LexA and QF lines that they generated can induce RNAi phenotypes when paired with LexAop or QUAS shRNA lines. In general, the claims of the paper are well supported by the evidence and the authors do a thorough job of validating the transgenic lines and characterizing their expression patterns.Strengths:Numerous Gal4 lines allow for highly specific genetic manipulation in a wide range of organs and tissues, however, similar tissue-specific drivers using alternative binary expression systems are not currently well developed. This study provides a large number of tissue and organ-specific LexA and QF2 driver lines that should be broadly useful for the *Drosophila* community.While a minority of the driver lines do not express the expected pattern (likely due to cryptic regulatory elements in the LexA or QF2 sequences), the ability to generate drivers using two different Gal4 alternatives mitigates this issue (as in nearly all cases at least one of the two systems produces a clean driver line with the expected expression pattern).The use of LexA-GAD provides an additional degree of control as it is subject to Gal80 repression. This could prove to be particularly useful in cases where a researcher wishes to manipulate multiple genes using Gal4 and LexA-GAD drivers as the Gal80(ts) system could be used for simultaneous temporal control of both constructs.The use of Fly Cell Atlas information to generate novel oenocyte-specific driver lines provides a useful proof-of-concept for constructing additional highly tissue-specific drivers.Weaknesses:Since these reagents will most commonly be paired with existing Gal4 lines, adding information about corresponding Gal4 lines targeting these tissues and how faithfully the LexA and QF2 lines recapitulate these Gal4 patterns would be highly beneficial.

It is outside the scope of this paper to analyze the expression patterns of the corresponding publicly available Gal4 lines. It is clear from the tissue specificity of the LexA-GAD and QF2 lines that they are expressed in the expected larval tissues based on the target genes. We have added a sentence in the discussion section noting “Further, we expect that there will also be differences between the expression pattern of corresponding Gal4 and the LexA-GAD/QF lines, as the latter were made by knock-in, while the former are often enhancer traps. However, based on our larval mounts and dissections, the stocks generated in this paper are highly specific to the expression pattern of the targeted genes.”

It is not stated in the manuscript if these transgenic lines and plasmids are currently publicly available. Information about how to obtain these reagents through Bloomington, Addgene, or TRiP should be added to the manuscript.

We have added to the materials section that “All vectors described here that are required to produce new driver lines will be made available at Addgene.” And “All transgenic fly stocks described here will be made available at the Bloomington *Drosophila* Stock Center.”

**Reviewer #2 (Public Review):**
Zirin, Jusiak, and Lopes et al presented an efficient pipeline for making LexA-GAD and QF2 drivers. The tools can be combined with a large collection of existing GAL4 drivers for a dual genetic control of two cell populations. This is essential when studying inter-organ communications since most of the current genetic drivers are biased toward the expression of the central nervous system. In this manuscript, the authors described the methodology for efficiently generating T2A-LexA-GAD and T2A-QF2 knock-ins by CRISPR, targeting a number of genes with known tissue-specific expression patterns. The authors then validated and compared the expression of double as well as single drivers and found the tissue-specific expression results were largely consistent as expected. Finally, a collection of plasmids for LexA-GAD and QF,2 as well as the corresponding LexAop and QUAS plasmids were generated to facilitate the expansion of these tool kits. In general, this study will be of considerable interest to the fly community and the resources can be readily generalized to make drivers for other genes. I believe this toolkit will have a significant, immediate impact on the fly community.
**Recommendations for the authors:**

**Reviewer #1 (Recommendations For The Authors):**
Lines 56-57: Janelia Flylight lines are not necessarily brain-specific - this collection has or could be screened in other tissues.

Correct. We have altered this sentence to read: However, these lines were developed primarily for brain expression. Although they are often expressed in other tissues, they are not well suited for experiments targeting non-neuronal cell types

Line 197 - I don't see the referenced Figure S1 in the reviewer materials. It appears this is actually referencing panels LL and MM in Figure 2.

Correct. We have fixed this error.

No information on the injection efficiency to create the CRISPR knock-in lines is presented. I am guessing the efficiency will be similar to that of other reported HDR-based CRISPR knock-ins, but if this information is available it would be useful to include it so that others know what to expect when injecting these vectors.

We did not systematically assay the injection efficiency. However, we can say that it was in line with previous descriptions of CRISPR-based plasmid and ‘drop-in’ HDR methods. We have added a note in the methods that “Knock-in efficiencies were comparable to previous reports (Kanca et al. 2019; Kanca et al. 2022).”

Demonstration of successful multi-manipulation would strengthen the paper.

We do not feel that this is necessary as there have been many papers showing combinatorial Gal4+LexA/QF experiments. An example from our lab can be seen in PMID: 37582831.

Also, are there approaches for efficiently constructing pairs of UAS/LexAOp or UAS/QUAS shRNA lines that would potentially streamline the genetics for multi-manipulation? Otherwise, this could be rather cumbersome to implement as one needs to combine a Gal4 line, a LexA/QF2 line (which will be constrained as to its chromosomal location by the target gene), and separate UAS-shRNA and LexAop/QUAS-shRNA constructs into the same fly.

There are some recent innovations that are useful in this respect. We have added a sentence to the discussion that says: “There remains an unmet need for a single vector that would allow for UAS/LexAop/QUAS control of different shRNAs. However, recent innovations in multi module vectors and multiplexed drug-based genetics allow researchers to more efficiently generate UAS/QUAS/lexAop transgenic fly strains (Matinyan et al. 2021; Wendler et al. 2022).”

In Figure 5 - is the difference for the hh inserts attributable to the driver line or the GFP/mCherry construct (or differential ability to detect GFP/mCherry)? One could try visualizing hhL(-Q) with the LexAop-GFP line. I guess that the correspondence between the nubbin and hh result suggests that maybe QF2 is suppressed in the wing pouch, but this could also be the difference in the reporter constructs and it would be interesting to know if this difference is truly attributable to the driver constructs from the standpoint of knowing how consistent the QF/LexA patterns are expected to be.

The difference is not attributable to GFP versus mCherry or the specific LexAop and QUAS lines that we used in figure 5. We tested the double knock-in and derivative single knock-ins with various QUAS and lexAop reporters and always observed the same pattern.

**Reviewer #2 (Recommendations For The Authors):**
There are a few points that should be clarified. A list of these specific points is provided below with the view that this could help the preparations of a stronger, improved paper.Line 50-51: "There have been no systematic studies comparing the two systems, with only anecdotal evidence to support one system over the other." It is unclear to me what the anecdotal evidence the authors referred to. Could the authors elaborate more on this part?

Based on an examination of QUAS brains, Potter et al, 2010 (PMID 20434990) makes the claim that “The low basal expression of QUAS and UAS reporters provides significant advantage compared to the lexA binary expression system.”

Shearin et al., 2014 (PMID: 24451596) compared Gal4/UAS, LexA/LexAop, and QF/QUAS reporter strength with the nompC driver and found that the QF system produced the strongest expression.

While these observations might be true in the nervous system, it isn’t clear that this extends to other tissues, nor what effect this would have on gene knockdown experiments.

There have been some reports that have explored swapping out a Gal4 insertion for a LexA or QF at the same locus. For example, Gohl et al. 2011 PMID: (PMID 21473015) mentions that “the majority of the swaps captured most features of the original GAL4 expression patterns. In some cases, however, either prominent features of the GAL4 pattern were lost or we observed new expression patterns. These changes may have resulted from differences in the strength or responsiveness of reporter lines. Alternately, the swap may have modified some combination of enhancer spacing and sequence composition flanking the promoter.”

Line 61-62: "On average, each StanEx line expresses LexA activity in five distinct cell types, with only one line showing expression in just one tissue..." What's the evidence to support this claim?

This observation comes from Figure S3 of Kockel et al. 2016 (PMID: 27527793), where the authors “analyzed a subset of 76 StanEx lines that are unambiguously inserted within, or adjacent to, a single known gene.” We cited this reference in the preceding sentence. To clarify, we have added the citation again for line 61-62.

Line 63-65: "These findings are consistent with prior studies indicating that enhancers very rarely produce expression patterns that are limited to a single cell type in a complex organism (Jenett et al. 2012)." It might be worth expanding on the use of the split system to achieve high cell-type-specificity. Especially, there are growing resources using split-intein and T2A-split-GAL4 with the prediction of genes from single-cell RNA sequencing datasets.

We agree that the split system is currently the premier method to produce the most specific driver lines. Indeed, our group has recently published a paper on the split-intein Gal4 system (see PMID 37276389). However, the tradeoff is that split systems usually require generation of transgenic lines, which becomes impractical for research involving two independent binary transcriptional systems, as the user would need to combine at least three driver components into single stocks, plus the UAS/QUAS/LexAop insertions. The ideal would be to generate complementary split insertions on the same chromosome, but we think a discussion of this is tangential to the thrust of our work here.

The authors did not fully discuss the rationale of using LexA-GAD vs LexA-p65 or VP16AD throughout the manuscript. I assumed the main reason for choosing LexA-GAD was to be compatible with GAL80 suppression. It might be worth explicitly stating in the result (e.g., line 123 or in the introduction). Also, did the authors observe weak transcriptional activation using LexA-GAD? It has been shown that the strength of transactional activation is much weaker for GAL4AD than the p65 or VP16AD. This might be worth noting in the manuscript as well.

We did briefly mention in the introduction that one disadvantage of the Flylight lines is that they “use a p65 transcriptional activation domain and therefore are not compatible with the Gal80 temperature sensitive Gal4 repression system.” We have expanded on this issue in the introduction which now says: “We chose to use LexA with the Gal4 activation domain, rather than the p65 or VP16 activation domains to allow for temporal control by Gal80 (Lai and Lee 2006; Pfeiffer et al. 2010). We chose to use QF2 variant over the original QF, to avoid the toxicity reported for the latter (Riabinina et al. 2015).”

We did not have any problems visualizing gene expression with fluorescent reporters. Nor did we have any difficulty obtaining knock-down phenotypes with ubiquitous drivers.

Line 125-127. Is there a specific reason why the authors chose the SV40 terminator for the double driver construct but the Hsp70 terminator for the single driver construct?

We found that the Hsp70 terminator gave slightly lower expression and decided to use this for the singles to avoid toxicity. For the doubles we chose the SV40, to compensate for reduced protein expressiojn of the second gene position.

Line 144-146: "To verify the knock-ins, we PCR-amplified the genomic regions flanking the insertion sites and confirmed that the insertions were seamless and in-frame." Did the authors recover lines with indel introduced, resulting in out-of-frame insertion?

Yes, we did see indels, which sometimes resulted in out of frame insertions, which were discarded. This result is in line with what we have observed with other CRISPR HDR knock-in experiments.

The underlying reason might be out of the scope of this manuscript. However, it would still be helpful for the authors to speculate the potential reasons why the T2A-LexA-GAD and T2A-QF2 targeting the same insertion site showed very distinct expressions.

It is outside the scope of this report to test this issue experimentally. We have a section in the discussion which does speculate as to the reason: “While we had no difficulty obtaining knock-ins for both types of activators, we did observe that for some target genes, the T2A-QF2 was only active in a subset of the expected gene expression pattern. In particular, we found that T2A-QF2 was difficult to express in the wing pouch. It may be that toxicity is an issue, and the weaker QF2w may be a better option for generating drivers in some organs (Riabinina and Potter 2016). Alternatively, differences in the LexA-GAD and QF2 sequences, and sequence length, could impact the function of nearby gene regulatory regions.”

Regarding the observation that the existence of 3XP3-RFP marker can interfere with the expression of T2A-LexA-GAD and T2A-QF2 expression in a case-by-case manner, it might be worth emphasizing in the discussion that the proper removal of 3XP3-RFP marker by Cre/LoxP recombination is important.

We have added the following to the discussion: “Importantly, our knock-in constructs contain the 3XP3-RFP cassette for screening transformants. Perhaps due to interaction between the 3XP3 promoter and the regulatory regions of the target gene, we occasionally saw misexpression of the LexA-GAD/QF2 in the 3XP3 domain. We have therefore prioritized Cre-Lox removal of the 3XP3-RFP cassette from our knock-in stocks, and advise that users of the plasmids described here likewise remove the marker, following successful knock-in.”

For Fig. 5B, 5F-G, the authors should elaborate more in the result section. For example, lines 215-217:"We tested this with the hh and dpp lines and observed robust generation of both T2A-QF2 and T2A-LexA-GAD from hs-Flp; T2A-QF2-T2A-LexA-GAD parents (Figure 5B)." It is unclear what the authors mean by "robust generation". Also, there is no description of the results in Fig. 5F-G.

We have expanded this section for figure 5B, which now reads: “We tested this with the hh and dpp lines and observed robust generation of both T2A-QF2 and T2A-LexA-GAD from hs-Flp; T2A-QF2-T2A-LexA-GAD parents (Figure 5B). In the case of the hh line, 15 out of 36 heat-shocked parents gave rise to at least one T2A-LexA-GAD progeny, with a mean of 14% recombinant offspring per parent. 20 out of 36 gave rise to at least one T2A-QF2 progeny, with a mean of 9% recombinant offspring per parent. In the case of the dpp line, 31 out of 32 heat-shocked parents gave rise to at least one T2A-LexA-GAD progeny, with a mean of 30% recombinant offspring per parent. 17 out of 32 gave rise to at least one T2A-QF2 progeny, with a mean of 9% recombinant offspring per parent.

We have also added a description for Figure 5F-G, which reads: “Recombinants were also independently verified by PCR of the insertions (Figure 5F-G), where we observed the expected smaller band sizes in the derivative T2A-QF2 and T2A-LexA-GAD relative to the parental double driver.”

Line 229, minor error: "Into these vectors, ..."

We have edited this to read: “We cloned shRNAs targeting forked (f) and ebony (e) genes into these vectors and assayed their phenotypes when crossed to ubiquitous LexA-GAD and QF2 drivers.”

Line 238-240: "Both Tub-LexA-GAD and Tub-QF2 drivers generated knockdown phenotypes in the thorax when crossed to f and e shRNA lines. However, the Tub-LexA-GAD phenotypes were stronger than those of Tub-QF2 (Figure 6C-D, F-G, I-J)." The stated "stronger phenotypes" are not clear to me. It might be worth elaborating more.

We have further clarified this by changing it to: “However, the Tub-LexA-GAD phenotypes were stronger than those of Tub-QF2 (Figure 6C-D, F-G, I-J). For example, Tub-LexA-GAD produced a fully penetrant f bristle phenotype (Figure 6F) while some wild-type bristles remained on the thoraces of Tub-QF2 f knockdown (Figure 6G). Neither Tub-LexA-GAD or Tub-QF2 was able to achieve the strength of phenotype generated by the T2A-LexA-GAD da knock-in line (compare the darkness of the cuticle caused by e knockdown in Figure 6H-J).”

Line 257-250: "Our collection of T2A-LexA-GAD and T2A-QF2 and double driver vectors can be easily adapted to target any gene for CRISPR knock-in, with a high probability that the resulting line will accurately reflect the expression of the endogenous locus" The authors could refer to the recent gene-specific Trojan GAL4/split-GAL4 work to support the idea that these gene-specific T2A-GAL4/split-GAL4 drivers reflect better than the enhancer-based drivers.

We have added the following sentence to the discussion: “The specificity achieved with this approach can also be seen in recent efforts to build collections of gene specific T2A-Split-Gal4 and T2A-Gal4 insertions (Kanca et al. 2019; Chen et al. 2023; Ewen-Campen et al. 2023).”

Line 630: "Removal of 3XP3-RFP eliminated gut and anal pad misexpression and did not affect glial cell expression." It would be helpful to add the annotation on Fig. 3B to show the location of glial cell expression.

We have added arrowheads on Figure 3 and the legend now reads: “Removal of 3XP3-RFP eliminated gut and anal pad misexpression and did not affect glial cell expression (white arrowheads).

Line 650-651: "The fat body mCherry expression is also present in the reporter stock and does not indicate LexA-GAD activity." I did not get what the authors were trying to convey. Where did the fat body mCherry expression come from? Please elaborate more.

We have changed this section to explain that “The fat body mCherry expression (yellow arrowhead) is from leakiness of the reporter stock and does not indicate LexA-GAD activity.”

Line 679-680: "forked shRNA produced a forked bristles phenotype." Please add the annotation on the figures to show where the phenotypes were.

We have added arrowheads and asterisks to the figure. The legend now reads: “(E-G) forked shRNA produced a forked bristles phenotype (white arrowheads). Note that some bristles retain a more elongated wild-type morphology with the Tub-QF2 driven forked knockdown (G, yellow asterisk).”

Fig 1D-E and 4A-B. There is no description throughout the manuscript about QA, QS regulation as well as little GAL80ts regulation. It will confuse readers with a little fly genetic background. Please include the introductions of these regulations of different binary expression systems.

We have added a section in the introduction, which states: “We chose to use LexA with the Gal4 activation domain, rather than the p65 or VP16 activation domains to allow for temporal control by the temperature sensitive Gal4 repressor, Gal80 (Lai and Lee 2006; Pfeiffer et al. 2010). We chose to use QF2 variant over the original QF, to avoid the toxicity reported for the latter (Riabinina et al. 2015). Like Gal80-based modulation of LexA-GAD, QF2 activity can also be regulated temporally by expressing QS, a QF repressor. QS repression of QF can be released by feeding flies quinic acid (Riabinina and Potter 2016).”

Fig. 2, there are several ND in the figure without any explanation in the manuscript (e.g. Mef2 and He). In addition, the expression patterns look quite different between T2A-LexA-GAD and T2A-QF2 for some genes (e.g., mex1, Myo31DF), but the authors did not mention any of them in the manuscript. Please elaborate more.

We have altered the Figure 2 legend as follows: “(A-KK) T2A-LexA-GAD knock-in lines crossed to a LexAop-GFP reporter and T2A-QF2 knock-in lines crossed to a QUAS-GFP reporter. Panels show 3rd instar larva. GFP shows the driver line expression pattern. RFP shows the 3XP3 transformation marker, which labels the posterior gut and anal pads of the larva. Gene names and tissues are on the left. We failed to obtain LexA-GAD knock-ins for Mef2 (E) and He (DD). (LL-MM) 3rd instar imaginal disc from the insertions in the nubbin (nub) gene. Note that most of the lines are highly tissue-specific and are comparable between the LexA-GAD and QF2 knock-ins. Insertions in the daughterless gene (da) and nub are an exception, as the T2A-LexA-GAD, but not the T2A-QF2, gives the expected expression pattern. Insertions in the gut-specific genes mex1 (X-Y) and Myo31Df (Z-AA) also differed between the LexA-GAD and QF2 drivers.”

We have also added a note on the inconsistency of mex1 and Myo31Df in the discussion: “While we had no difficulty obtaining knock-ins for both types of activators, we did observe that for some target genes, the T2A-QF2 was only active in a subset of the expected gene expression pattern. In particular, we found that T2A-QF2 was difficult to express in the wing pouch. Additionally, we found that the driver expression in the gut-specific genes, mex1 and Myo31Df differed between the LexA-GAD and QF2 transformants. In both cases the LexA-GAD was more broadly expressed along the length of the gut than the QF2. It may be that toxicity is an issue, and the weaker QF2w may be a better option for generating drivers in some organs (Riabinina and Potter 2016).”

Fig. 4B, it is unclear why the hsp70 is present downstream of the enhancer of interest (upstream of T2A). Is it the molecular mark resulting from the cloning steps? Does it serve any specific purpose?

This is the *Drosophila* hsp70 gene minimal promoter and is standard for many expression constructs in *Drosophila*. In the methods section we described how we made versions of the pMCS-T2A-QF2-T2A-LexA-GAD-WALIUM20 with and without tis minimal promoter: “We used pMCS-T2A-QF2-T2A-lexA0GAD-WALIUM20 for dpp-blk and pMCS-T2A-QF2-T2A-lexGAD-WALIUM20-alt (which lacks the hsp70 promoter) for Ilp2, since dpp-blk does not have a basal promoter, but the Ilp2 enhancer does.”

Fig 5A. The resulting single T2A-QF2 and T2A LexA-GAD from the double driver parental lines retain the sequence of FRT3 upstream of the QF2 and LexA-GAD. I assume the FRT3 part will be translated and remain attached to QF2 and LexA-GAD. Is that correct? If so, would this cause any adverse effect?

Correct. The FRT3 sequence is present in both the parental double and single derivatives. We can say that the additional amino acids do not prevent LexA-GAD or QF2 transcriptional activation. We do not know whether there may be other adverse effects, though we did not observe any.

Fig. 5C-C'. It seems like the images of Fig. 5C-C' were the same as Fig. 4D-D'. If so, the authors should indicate that in the figure legend.

We have made a note of this in the figure legend.